# Impacts of Agricultural Capitalization on Regional Paddy Field Change: A Production-Factor Substitution Perspective

**DOI:** 10.3390/ijerph18041729

**Published:** 2021-02-10

**Authors:** Xinyi Li, Xiong Wang, Xiaoqing Song

**Affiliations:** 1Research Center for Spatial Planning and Human-Environment System Simulation, School of Geography and Information Engineering, China University of Geosciences, Wuhan 430074, China; lxy156@cug.edu.cn (X.L.); wxrwx@cug.edu.cn (X.W.); 2Hunan Key Laboratory of Land Resources Evaluation and Utilization, Hunan Planning Institute of Land and Resources, Changsha 410007, China

**Keywords:** paddy field, labor productivity, agricultural capitalization, production-factor substitution, China

## Abstract

Paddy fields are significant in ensuring food security and improving the agricultural ecological environment. In economic terms, paddy field use is affected by input costs and crop market price. There is insufficient understanding of factor input costs caused by agricultural production-factor substitution, driving paddy field change. This study uses a panel regression model to analyze the influence of agricultural production-factor substitution on paddy field use from 1990 to 2016. The case area is Hubei province, China. The results show that the overall growth trend in paddy fields is unequivocal in China’s grain production areas. The improvement in agricultural production conditions, including irrigation and land quality, has a positive effect on the area proportion of paddy fields. With socioeconomic developments, the relationship between the substitution of nitrogen fertilizer for farmland and the area proportion of paddy field is inverted-U shaped, while the effect of the substitution of machinery for labor is U-shaped. The main conclusion is that the process of agricultural production-factor substitution, intended to maximize labor and land productivity, will increase the area proportion of paddy field. Public policies should focus on improving the level of agricultural mechanization and crop diversity to protect food security and sustainable agricultural intensification.

## 1. Introduction

At the end of the 1990s, the global market demand for cash crops increased sharply, and the planting area of cash crops, such as, vegetables expanded rapidly. Many paddy fields were converted into dry land. This was the main form of internal change in global farmland use [1,2]. The ecosystem service functions of paddy fields are significantly better than those of dry land in terms of climate regulation, flood storage, pollution removal, water loss prevention, biodiversity conservation, etc. [3,4,5]. When paddy fields are converted into dry land, the change in the planting structure of farmland causes increased fertilizer and pesticide use, planting intensity, and agricultural water consumption. These factors have a negative influence on the regional ecological environment [6,7,8]. Therefore, to protect food security and improve farmland ecosystems, it is important to understand the evolution and driving mechanisms of paddy field change.

The relevant literature focuses on the dynamics of paddy field changes, concentrating on Southeast Asia. For example, based on remote sensing image data, Ichinose et al. [9] pointed out that the area proportion of paddy fields in Akashima, central Japan, declined from 57.3% to 44.3% between 1963 and 2000. Liu et al. [10] found out that the area occupied by paddy fields in China has decreased significantly since the 1980s. Nahib et al. [11] showed that residential and construction land in the Indramayu Regency of Indonesia reduced the area of paddy fields from 54,961 ha in 1990 to 13,006 ha in 2011. The rapid decrease in the area of paddy fields in Southeast Asia has a profound influence on regional food security and environmental issues [12,13]. An in-depth analysis of the driving mechanisms of paddy field changes could optimize regional farmland use. Farmland use change is influenced by natural, economic, social, and other factors, as is paddy field change [14,15]. Scholars have explored the driving forces of paddy field loss and they agree that the expansion of construction land and the increase in dry land area are the main reasons for the decrease in paddy fields [16,17].

We found a paddy field expansion trend in some parts of the world. There has been an expansion of paddy fields in Africa, caused by an increase in food consumption [18]. Panichvejsunti et al. [19] showed that after Thailand implemented its rice pledge policy, the increase in rice purchase prices caused an expansion of the rice planting areas. Also, large areas of dry land were converted into paddy fields in Northeast China because rice had a unique price advantage in this region [20,21]. In the context of the expansion of market demand for cash crops, the area of paddy fields does not necessarily decrease. It is possible to protect food security and the agricultural ecological environment by preventing the conversion of paddy fields to dry land. What is the driving mechanism for the changing extent of paddy field areas? Studies in this vein have mainly been carried out from a market perspective [22,23]. Changes in paddy field areas are affected by various factors such as the market price and input factor costs. However, there have been few studies on the influence of production-factor change on paddy field change.

At present, global urbanization is developing rapidly. In developing countries, phenomena occur, such as construction land expansion into farmland and the removal of agricultural workers from agricultural production to work in the cities [24,25,26]. Socioeconomic developments have caused severe farmland loss and labor transfer, and the increase in capital inputs such as machinery and fertilizers has become the main driving force for agricultural economic growth. The global agricultural production-factor input structure has undergone significant changes [27,28,29,30]. Especially in China, a country with the fastest urbanization rate in the world, the direct integration of farmers and production materials was realized under the Household Contract Responsibility System in 1978 [31]. The development of urbanization and industrialization caused farmland loss and labor transfer. Simultaneously, agricultural intensification and large-scale management advanced rapidly, promoting fertilizer, machinery, and other capital factors replacing land and labor factors [32,33,34], which played a key role in improving grain production [35,36,37]. Previous research has shown that the use of capital factors to replace land and labor factors has a significant impact on planting structure [38,39,40]. Still, there are few studies on the driving mechanisms of regional paddy field changes based on the perspective of production-factor substitution. Also, many researchers have pointed to the socioeconomic transformation of 20th century China, from a low level of development to an accelerated development stage [41,42]. In this period, the Lewis turning point in agricultural labor supply in China has been reached [43,44]. Labor supply changed from surplus to scarcity. Farmland resources experienced a qualitative change in the same period, and the farmland protection policy resulted in a rise after a rapid decline [24,45]. Many researchers have shown that the influence of socioeconomic development on production-factor inputs has typical nonlinear characteristics [46,47]. Therefore, we believe that the nonlinear characteristics of agricultural production-factor inputs will lead to a nonlinear relationship between agricultural production-factor substitution and paddy field change. The specific hypotheses are:

**Hypothesis** **1.**
*The substitution of nitrogen fertilizer for farmland had a nonlinear impact on paddy field change during the study period.*


**Hypothesis** **2.**
*The substitution of machinery for labor had a nonlinear impact on paddy field change during the study period.*


The purpose of this study was to evaluate the influence of agricultural production-factor substitution on paddy field change by using a panel regression model. Policy suggestions were put forward to safeguard food security and agricultural sustainable intensification. This study is structured as follows: Section 2 presents the materials and methods. Section 3 presents the results. Section 4 presents the discussion. Section 5 presents the conclusions. Our results will enhance the understanding of the driving mechanism of paddy field change in China’s grain production areas. They will serve as a scientific reference for the improvement of food security and agricultural sustainable intensification in China.

## 2. Materials and Methods

### 2.1. Study Area

Hubei province is located in the middle reaches of the Yangtze River, central China (Figure 1a). This province is dominated by plains in the middle, with surrounding mountains in the west, and hills in the east (Figure 1b,c). The relief degree of land surface (RDLS) of each city in Figure 1c was extracted by ArcGIS 10.2 on the basis of 90m Digital Elevation Model (DEM)data in Figure 1b. The total area is 185,900,00 ha. Hubei province has a subtropical monsoon climate with abundant agricultural natural resources. The annual average temperature ranges between 15 °C and 17 °C, the frost-free period ranges between 230 days and 307 days, and the average annual precipitation ranges between 850 mm and 1700 mm, according to data available from Baidu Baike, China’s largest Internet search engine, (https://baike.baidu.com/ (accessed on 9 February 2021)). It is the superior natural conditions that make Hubei province an important grain production area and a production base for major cash crops such as cotton, oil, and vegetables. However, from 1990 to 2016, the area of paddy field in Hubei province increased from 1,871,780 ha to 2,026,670 ha according to the Rural Statistical Yearbook of Hubei Province. The approximate distribution of paddy fields is shown in Figure 1d. The increase in paddy field area has a positive effect on maintaining regional food security and improving agricultural ecosystem service values. An in-depth analysis of the driving mechanisms of paddy field changes in Hubei province will be of great significance to other regions. Therefore, it was appropriate to take this province as the study area to analyze the driving mechanism of factor substitution on paddy field change.

### 2.2. Data

This paper used data which included climate data, socioeconomic data, farmland data, and agricultural production data from 1990 to 2016. The climate data include average annual temperature and average annual precipitation, which is available from the National Meteorological Data Center (http://data.cma.cn/ (accessed on 9 February 2021)). The socioeconomic data include per capita GDP and net income of farmers, from the Hubei Statistical Yearbook. The farmland data are taken from the Hubei Rural Statistical Yearbook. Agricultural production data include effective irrigated area, grain sown area, total grain output, total crop sown area, agricultural employees and rural labor force, total power of agricultural machinery, the net amount of nitrogenous fertilizer input, and specific crop sown area (including early rice, mid-season rice (including one season late), double-season late rice, wheat, silkworm pea, corn, sorghum, soybean, potatoes, other grains, peanut, rapeseed, sesame, cotton, raw jute, raw ramie, sugar, tobacco, Chinese medicine crops, vegetables (including edible fungi), melons and fruit, green fodder, green manure, and other ecological crops, a total of 23 kinds of crops), all from the Hubei Rural Statistical Yearbook.

### 2.3. Methodology

#### 2.3.1. Index Selection

Dependent variable: It is inevitable that the construction land expansion will occupy the paddy field. Preventing the conversion of paddy fields into dry land is of positive significance in protecting food security and the agricultural ecological environment. Therefore, proportion of paddy field area (*PPF*) was a suitable dependent variable.

Core independent variables: Socioeconomic development has led to farmland loss and labor transfer. The widespread use of nitrogen fertilizer and machinery in agricultural production replaces farmland and labor, respectively. Therefore, agricultural machinery per capita of agricultural employees (*AMAE*) and the input intensity of nitrogen fertilizer per hectare of farmland (*INNF*) were introduced as the core independent variables in the model. To explore the nonlinear characteristics of the substitution of agricultural production factors on paddy field change, we introduced the quadratic term of *AMAE* (*AMAE^2^*) and *INNF* (*INNF^2^*) to validate the nonlinear effect in the model.

Control variables: Apart from the core independent variables, the model should also add some control variables to reflect the factors affecting paddy field changes. These control variables should include climate change, agricultural production conditions, farmers’ risk awareness, and socioeconomic development. We selected the average annual temperature (*AT*) and average annual precipitation (*AP*) to represent climate change [48,49,50]. In terms of the influence factors of agricultural production conditions, we chose effectively irrigated farmland (*PEIF*) and land productivity (*LP*) to represent farmland water conservancy facilities and farmland quality [51,52]. Crop diversity (*CD*), which means the crop evenness [53,54], was chosen to represent farmers’ risk awareness [55]. In terms of the socioeconomic factors, the proportion of the number of non-agricultural employees (*PNAE*), proportion of cash crop sown area (*PCC*), and income inequality (*II*) were introduced into the model as control variables [56,57]. The independent variables are listed in Table 1 and descriptive statistics covering all variables are shown in Table 2.

#### 2.3.2. Regression Model

Panel data have a number of advantages, including facilitating the control of individual heterogeneity, reducing the collinearity among variables, and increasing degrees of freedom and effectiveness. In order to eliminate collinearity, all variables are processed logarithmically before regression. Because the time series was long, where T was more than 15, it was necessary to carry out a unit root test. In order to avoid the shortcomings and deficiencies of a single method, five unit root tests were employed: the Breitung test, Levine–Lin–Chu test, Im, Pesaran and Shin test, Fisher Augmented Dickey–Fuller test, and the Fisher–Phillips–Perron test [58]. According to the principle that the minority should subordinate to the majority, *logPPF*, *logPEIF*, and *logAMAE* were first-order single integrations, while the rest were zero-order single integrations which belonged to an unbalanced panel, so a co-integration test is required. The *p*-values of Kao testwas 0.00 < 0.01, thus rejecting the null hypothesis, meaning that there was a co-integrating relationship [59], and the next step of regression analysis could be carried out. There are variable intercept models and variable coefficient models in panel regression. This paper aims to investigate the driving mechanism of paddy field use change in Hubei province from 17 prefecture-level cities, without obtaining specific regression results for each such city, so the variable intercept model was adopted. To determine whether the data were amenable to fixed-or random-effects modeling, a Hausman test was employed [60]. The result showed that *p* = 0.00 < 0.05, which meant that the fixed-effects model was more appropriate. Therefore, the equation of the regression model is as follows:(1)$$lnY_{it}=\alpha_{i}+y_{t}+\beta +\beta_{1}lnX_{1it}+\beta_{2}lnX_{2it}+\beta_{3}lnX_{3it}+\beta_{4}lnX_{4it}+\beta_{5}lnX_{5it}+\beta_{6}lnX_{6it}+\beta_{7}lnX_{7it}+\beta_{8}lnX_{8it}\;+\beta_{9}lnX_{9it}+\beta_{10}X_{9it}^{2}+\beta_{11}lnX_{10it}+\beta_{12}X_{10it}^{2}+u_{it}$$
where $\alpha_{i}$ and $y_{t}$ represent cross-sectional and time-fixed effect factors respectively, while *β* denotes the common intercept, and $\;\beta_{1}$*,*$\;\beta_{2}$*,*
$\beta_{3}$*,*
$\beta_{4}$*,*
$\beta_{5}$*,*
$\beta_{6}$*,*
$\beta_{7}$*,*
$\beta_{8}$*,*
$\beta_{9}$*,*
$\beta_{10}$*,*
$\beta_{11}$*,*
$\beta_{12}$*,* denote the coefficients to be estimated, $\;X_{1}$*,*$\;X_{2}$*,*
$X_{3}$*,*
$X_{4}$*,*
$X_{5}$*,*
$X_{6}$*,*
$X_{7}$*,*
$X_{8}$*,*
$X_{9}$*,*
$X_{10}$ are *AP*, *AT*, *PEIF*, *LP*, *PCC*, *II*, *PNAE*, *CD*, *AMAE*, *INNF*.${\;u}_{it}\;$is the error term, *i* and *t* denote the region and year, respectively.

## 3. Results

### 3.1. Changes in Paddy Field Use and Its Influencing Factors

From 1990 to 2016, the *PPF* in Hubei province increased from 53.836% to 58.841%. According to Figure 2, at the urban scale *PPF* tends to exhibit an increasing trend, which means that there was a lot of dry land conversion to paddy fields in Hubei province. The number of cities where the area occupied by paddy fields is increasing over time is significantly higher than the number of cities where it is decreasing, accounting for more than 70% of the prefecture-level cities in the whole province. Specifically, the major regions with substantial growth are Ezhou, Jingzhou, and Xiaogan, where the increases were 22.191%, 14.063%, and 10.093%, respectively. The regions where the *PPF* decreased are Yichang, Xianning, Wuhan, Huangshi, and Enshi (3.708%, 3.325%, 1.643%, 0.414%, and 0.393%, respectively). The magnitudes of these decreases are clearly smaller than the magnitudes of the increases; areas experiencing declines in paddy fields are mainly distributed in Southwest and Southeast Hubei.

The value of *AT* and *AP* in Hubei province showed an obvious upward trend between 1990 and 2016 (Figure 3a). Specifically, values for *AT* increased from 16.339 °C to 16.885 °C, while *AP* increased from 1137.794 mm to 1421.432 mm. During the same period, the value of *PEIF* increased from 68.066% to 84.360%, and its growth rate was slow at first, and then fast. By contrast, the value of *LP* increased from 4908.700 kg/ha to 5489.963 kg/ha; its growth rate was fast first and then slow (Figure 3b). The value of *CD* in Hubei province decreased by 0.008, and its trend changed from upward to downward between 1990 and 2016 (Figure 3c). The values of *II, PNAE*, and *PCC* increased from 2.300 to 4.325, 18.832% to 62.301%, and 23.759% to 39.324%, respectively; all exhibited increases first and then declined (Figure 3d,e).

During the study period, values of *AMAE* increased from 0.756 kW to 4.850 kW, and the *INNF* increased from 262.325 kg/ha to 388.952 kg/ha. Both showed remarkable growth and significant phased difference. The growth rate of *AMAE* was slow at first and then fast, while the opposite is the case in terms of *INNF* (Figure 3f).

### 3.2. Panel Regression Results

Based on the panel data of 17 cities during 1990–2016, the panel regression model with a quadratic term was used to estimate the driving mechanism of paddy field change in Hubei province. The test results of the model are as follows (Table 3). Robustness checks are employed to confirm the validity of findings and the results are presented in Table A1 of the Appendix A. The coefficient direction and significance level of the core variables in robust test (Table A1 in the Appendix A) are consistent with the results of panel regression (Table 3). These results validate the nonlinear influence of agricultural production-factor substitution on paddy field change. Thus, our findings are robust by using different proxy and by altering the regression approach.

#### 3.2.1. Impact of Climate Change

Within a short period, climate change denoted by changes in *AT* and *AP* are not major driving factors for agricultural production. In the model, the *logAT* and *logAP* coefficients are not significant, indicating that climate change does not have a significant impact on paddy field use between 1990 and 2016 (Table 3).

#### 3.2.2. Impact of Agricultural Production Conditions

The *logLP* coefficient is 0.103 and is significant at the 10% level. This indicates that there is a significant positive relationship between *LP* and *PPF*. High-quality farmland is conducive to stabilizing crop yield and is usually utilized to produce crops suitable for large-scale and intensive management [61,62]. In Hubei province, rice is the most demanding and mechanized crop, and its cultivation in high-quality farmland can effectively reduce risks in agricultural production. Considering this, farmers tend to develop large-scale management approaches and obtain scale benefits. Therefore, the improvement of *LP* is of positive significance to *PPF* [63].

The *logPEIF* coefficient is 0.244 and significant at the 1% level. This implies a significant positive relationship between agricultural infrastructure and *PPF* (Table 3). This is because the optimization of water conservation provides sufficient water resources for farmland, which is beneficial to rice cultivation and the expansion of *PPF*. This result is consistent with the findings in cognate studies [64,65,66].

#### 3.2.3. Impact of Farmers’ Risk Consciousness

The *logCD* coefficient is 0.967 and is significant at the 1% level. This result shows that farmers’ risk consciousness is positively related to *PPF* (Table 3). Hubei province is located in a subtropical monsoon region characterized by sufficient rain and heat. Double cropping rice (including early rice and late rice) can be planted within a year. Studies have shown that double-season planting is an effective way to improve the planting area of paddy fields and the evenness of its planting structure [53]. Farmers with higher risk consciousness are more inclined to grow double cropping rice because they can continue to grow late-season rice to ensure income after flooding (frequent from late June to mid-July in Hubei province) [67]. Therefore, raising risk consciousness is a favorable factor for increasing *PPF*.

#### 3.2.4. Impact of Socioeconomic Development

The *logPCC* coefficient is −0.140 and is significant at the 1% level. This indicates a significant negative relationship between market benefits inducement and *PPF* (Table 3). High market benefits induce farmers to switch from planting traditional grain crops to cash crops, but most of the cash crops are dry land crops, which will inhibit *PPF*. This is consistent with the results of Su et al. [68], that current market benefits induce cash crops to replace grain crops and occupy traditional rice fields.

The *logII* coefficient is −0.142 and significant at the 1% level. This means that the expansion of the income gap has an inhibitory effect on *PPF* (Table 3). This is because the introduction of cash crop cultivation is the main way for farmers to increase household income and narrow the societal income gap [69].

The *logPNAE* coefficient is −0.110 and significant at the 1% level. This reveals that the transfer of labor to the non-agricultural sector will hinder *PPF* (Table 3). On the one hand, the loss of rural labor will promote land circulation; on the other hand, it will lead to the aging and feminization of the agricultural labor force, both of which will expand the sown area of cash crops [39,70].

#### 3.2.5. Impact of Agricultural Production-Factor Substitution

The *logAMAE* coefficient is −0.002 and is not statistically significant (Table 3). Related research has shown that the moderate scale of farmland per worker in Hubei province is 1.162 ha [71], however, it was only 0.399 ha in 2016. It is the low-scale level that the *AMAE* had no significant effect on *PPF*. The (*logAMAE*)^2^ coefficient is 0.026, and significant at the 1% level (Table 3). This indicates that there is a U-shaped relationship between the substitution of machinery for labor and *PPF*, which proceeds from a significant negative to a significant positive relationship. When the first farm machinery was put into use, they contributed to a decrease in *PPF*, but *PPF* increased as the substitution of machinery for labor became more widespread.

The *logINNF* coefficient is 0.405 and is significant at the 1% level (Table 3). This indicates a positive relationship between the substitution of fertilizer inputs for farmland inputs and *PPF* during the study period. Fertilizer inputs could effectively increase output, and such inputs in paddy fields can lead to higher benefits than dry land because the production and production potentiality per unit area of grain crops are higher than cash crops. The (*logINNF*)^2^ coefficient is −0.029 and is significant at the 1% level (Table 3). This implies an inverted-U shaped relationship between the substitution of nitrogenous fertilizer for farmland and *PPF*, proceeding from a significant positive to a significant negative relationship. When the first ton of nitrogen fertilizer was put into use, they contributed to an increase in *PPF*, but *PPF* dereased as the substitution of nitrogenous fertilizer for farmland became more widespread.

## 4. Discussion

In this paper, a panel regression model with a quadratic term was established to analyze the nonlinear influence of agricultural production-factor substitution on paddy field change during social development. This study fills a gap in the research on paddy field use change resulting from substituting agricultural production factors. The results correctly verify the hypothesis that agricultural production-factor substitution has a nonlinear effect on paddy field change.

### 4.1. Mechanisms of the Production-Factor Substitution on PPF

During the research period, even at different socioeconomic development stages, agricultural production-factor substitution had a positive effect on *PPF*. When the social economy is at a low-level developmental stage, traditional agriculture shows the characteristics of scarce farmland resources and sufficient labor resources, aiming at meeting the needs of family consumption. The low level of socioeconomic development limits the onset of large-scale substantive rural–urban migration; workers replaced by machinery were still engaged in agricultural activities [72]. Farmers prefer to grow cash crops, thereby creating more employment opportunities and economic benefits [73,74], and this limited *PPF* growth. Because the agricultural capital available to farmers was limited, farmers also tended to invest in fertilizer, rather than in machinery. Increasing nitrogen fertilizer input can enhance output and achieve the maximization of land productivity [75,76]. The limited agricultural capital was invested in planting rice, the most important survival crop when there was poverty among farmers [77]. Figure 4 shows that, in the early stage, the rate of farmland loss was significantly higher than that of labor loss. Land was the scarcest resource. The power of nitrogen fertilizer input was stronger than that of machinery input. The aim of agricultural production was to pursue the land productivity maximum. Therefore, the *INNF* was the first factor in the growth of *PPF* in the early stage.

During rapid socioeconomic development, many rural workers move to non-farm sectors, resulting in labor replacing land as the scarcest resource [66,78,79]. The main purpose of farming is no longer to satisfy the needs of family consumption but more to obtain economic benefits. The needs of family consumption can be met in the market. At this stage, farmers have more money to invest in farming. Capital profitability makes it more likely that fertilizer is applied on dry land areas with high-income potential, while machinery is used in paddy fields because rice planting is easier to upscale and more amenable to mechanized production than other cash crops [80,81]. Therefore, the *AMAE* and *INNF* had positive and negative effects on *PPF*, respectively. When a strict cultivated land protection policy was implemented in China, the total area of farmland increased (see Figure 4). Compared with seeking the land productivity maximum, farmers are more inclined to seek the labor productivity maximum. Therefore, the *AMAE* was the first factor in the growth of *PPF* at the later stage.

It can be concluded that with the change in the cost of production-factor, technological change has occurred in Hubei province, where farmland inputs have been replaced by fertilizer inputs and labor inputs by mechanical inputs. However, as the scarcity of agricultural production-factors and the management objectives differed in diverse socioeconomic development stage, and production-factor substitution has a nonlinear impact on *PPF*. In the early stage, the scarcity of land leads to *INNF* as the first factor of the growth of *PPF*, while in the later stage, the scarcity of labor leads to *AMAE* becoming the first factor of in *PPF* growth (see Figure 5). That is to say, the shift from the land productivity maximization to labor productivity maximization [82,83,84], caused by socioeconomic developments, has a positive effect on *PPF*.

### 4.2. Policy Implications for Food Security in China

The *PPF* in Hubei province increased significantly during the study period, which is not only conducive to maintaining regional food security, but also improving agricultural ecosystem service values. The development experience offers important lessons to other regions. However, it can be seen from Figure 5 that in different stages, the increase in *PPF* in Hubei provincehas occurred under certain conditions. Rice is the most demanded crop in Hubei province, and its long tradition of planting is a necessary condition for maximizing land productivity and promoting the increase in *PPF*. In addition, the high mechanization degree of rice planting is the main factor which explains the increase in *PPF* in the period of pursuing the maximization of labor productivity. Therefore, the consistent growth of *PPF* in Hubei province during the study period is inseparable from cultural traditions, natural conditions, and the high level of mechanization associated with local rice planting.

In the context of expanding market demand for cash crops, some farmers have changed from paddy field cultivation to dry land cultivation in order to pursue market income, resulting in declining eco-environment benefits of farmland. Therefore, this paper puts forward the following policy recommendations on how to increase *PPF* in regions with a long tradition of rice planting such as Hubei province.

(1) Agricultural production conditions should be optimized to increase *PPF*. The government should guide land consolidation to improve the irrigation capacity and quality of farmland [85], and also encourage land circulation to promote large-scale operations [86]. It should not only create favorable conditions for the promotion of agricultural mechanization, but also increase *PPF* to ensure regional food security and improve agricultural eco-environment benefits.

(2) Agricultural subsidy policies should be more oriented to farm machinery purchase subsidy [87,88]. The high labor productivity brought by machinery inputs is a key factor for the increase in *PPF*. For areas rich in paddy fields, raising the subsidy standard of agricultural mechanization planting is an effective measure to promote the increase in *PPF*.

(3) Differentiated regional agricultural policies should be implemented based on local conditions [89,90]. There are different mechanisms in the impact of factor substitution on paddy field use in different stages of social development. For regions with a low socioeconomic development level, fertilizer inputs can be appropriately increased to promote *PPF*; and for regions with a higher socioeconomic development level, mechanization should be improved to increase *PPF*.

### 4.3. Proximate Impacts Maximizing Labor Productivity on Sustainable Intensification

In recent decades, the rapid development of industrialization and urbanization has led to a large amount of farmland loss [91], the degradation of farmland quality [92,93,94], and environmental pollution [95,96,97,98]. The sustainable intensification of paddy field use plays an important role in agriculture sustainable intensification.

(1) The level of regional crop diversification has declined. Maximum labor productivity is achieved mainly through specialized production of a certain crop with a large amount of input in terms of machinery, fertilizer, and pesticides. When there are agricultural products with comparative advantages in a certain region, the pursuit of maximizing labor productivity will increase the sown area of those agricultural products. The declining level of crop diversity caused by specialized planting will lead to the simplification of regional agricultural ecological environments. We suggest that the government should guide farmers to grow a variety of crops by adjusting the crop market prices [99,100]. In addition, a diversified planting subsidy policy could be implemented to encourage farmers to adopt diversified planting [101,102].

(2) Farmland circulation and abandonment will coexist, which causes the polarization of farmland use. On the one hand, the pursuit of maximum labor productivity will promote land circulation and increase farmland use intensity [103]; on the other hand, the increase in farming costs will reduce farmers’ willingness to cultivate, and then lead to farmland degradation and abandonment [104,105]. Finally, the pressure of grain production is concentrated on less farmland. Therefore, we should actively promote land consolidation and land circulation, in order to alleviate the decline in grain production capacity and the destruction of the ecological environment caused by farmland abandonment.

(3) Agricultural production pays more attention to economic benefits rather than environmental benefits. Although high inputs of pesticides and fertilizers can increase food production, excessive inputs will reduce farmland ecosystem service functions, and even cause serious pollution [106]. Here, we suggest that the prices of pesticides and fertilizers should be appropriately increased to achieve the goal of reducing their input.

### 4.4. Limitation

In this study, we used a panel regression model with a quadratic term to estimate the nonlinear influence of agricultural production-factor substitution on paddy field change. This research method can identify the nonlinear influence, but it cannot identify the precise time of the turning point. This problem could be solved by increasing the sample size and by changing research methods. Also, our indicators could be improved if complete data could be extracted. In addition, the rapid urbanization of central China has made Hubei province a major grain-producing area. This study could be replicated in the economically developed grain-producing areas to explore the applicability of the research results of this paper.

## 5. Conclusions

This study used a panel regression model with a quadratic term to explore the nonlinear features of agricultural production-factor substitution in paddy fields’ change of use.

The overall growth trend in *PPF* in Hubei province during the period 1990 to 2016 is clear. The number of cities with increasing areas in paddy field is significantly higher than the areas where these fields are decreasing, and the rapid growth areas are concentrated in the central region.

The improvement in agricultural production conditions, including irrigation conditions and land quality, has a positive effect on the increase in *PPF*. However, socioeconomic factors, including the transfer of labor, the expansion of cash crop demands, and income inequality, have negative effects on *PPF*.

With socioeconomic developments, the relationship between *INNF* and *PPF* is inverted U-shaped, while the effect of *AMAE* is U-shaped. During this period, labor replaced land as the scarcest resource in agricultural production, and the first factor that agricultural production-factor substitution on paddy field change also changed from the *INNF* to *AMAE*. This process positively affects *PPF*.

## Figures and Tables

**Figure 1 ijerph-18-01729-f001:**
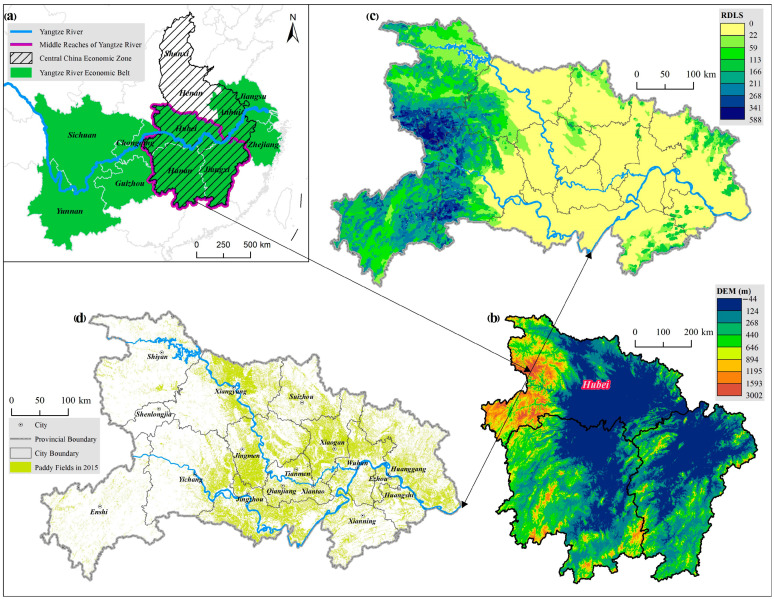
Location of (**a**) the middle reaches of the Yangtze River, (**b**) the Digital Elevation Model (DEM) of the middle reaches of the Yangtze River, (**c**) the relief degree of land surface (RDLS) of Hubei province, (**d**) the distribution of paddy fields in Hubei province in 2015. Data sources: the Data Center for Resources and Environmental Sciences, Chinese Academy of Sciences (RESDC) (http://www.resdc.cn (accessed on 9 February 2021)).

**Figure 2 ijerph-18-01729-f002:**
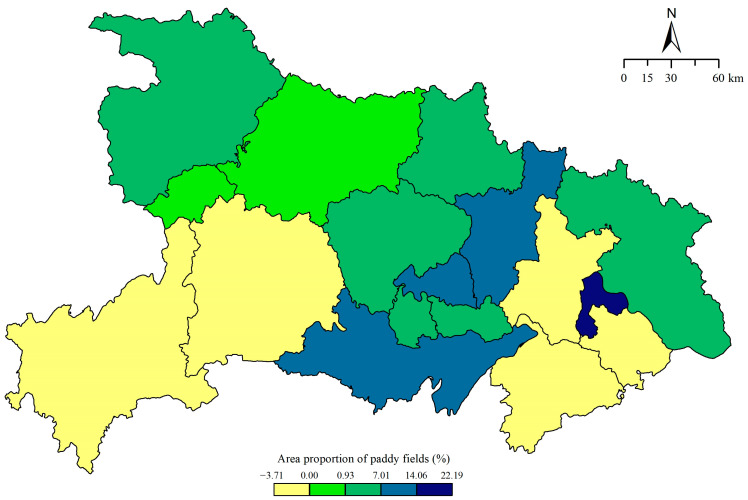
Spatial distribution of the area proportion of paddy fields variations in Hubei province from 1990 to 2016. Data source: Rural Statistical Yearbook of Hubei Province 1991 and 2017.

**Figure 3 ijerph-18-01729-f003:**
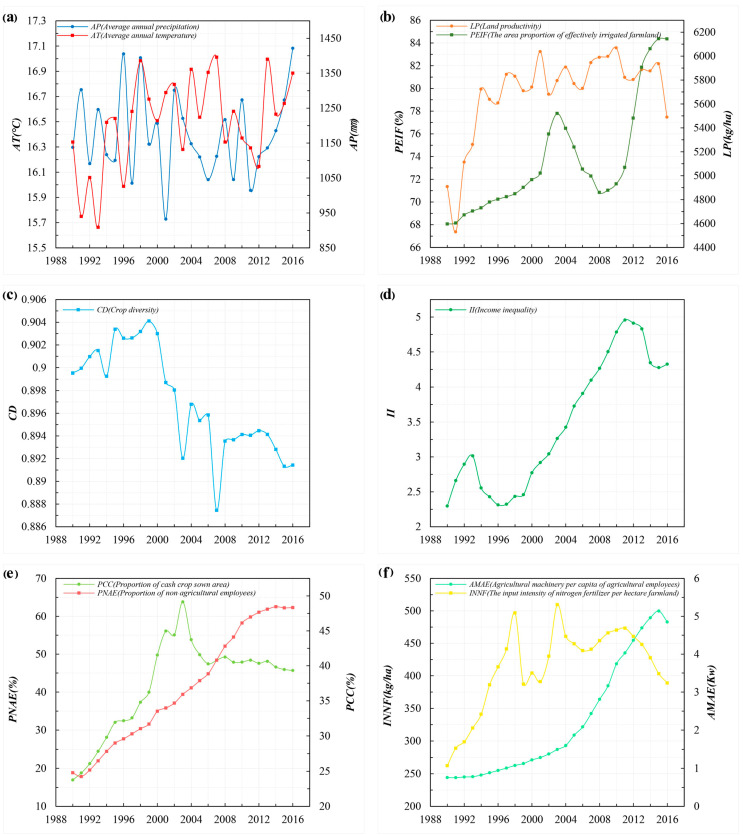
Changes in (**a**) average annual temperature (*AT*), average annual precipitation (*AP*), (**b**) proportion of effectively irrigated farmland area (*PEIF*), land productivity (*LP*) (**c**) crop diversity (*CD*) (**d**) income inequality (II) (**e**) proportion of cash crop sown area (*PCC*), proportion of the number of non-agricultural employees (*PNAE*) (**f**) agricultural machinery per capita of agricultural employees (*AMAE*), and input intensity of nitrogen fertilizer per hectare of farmland *(INNF*) within Hubei province between 1990 and 2016. Data source: The Rural Statistical Yearbook of Hubei Province 1991–2017 and the Hubei Statistical Yearbook 1991–2017.

**Figure 4 ijerph-18-01729-f004:**
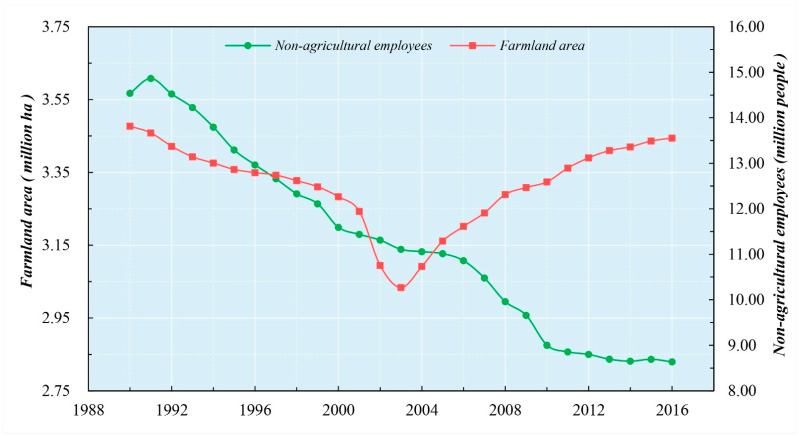
The driving mechanism of the area proportion of paddy fields within Hubei province between 1990 and 2016. Data source: the Rural Statistical Yearbook of Hubei Province 1991–2017.

**Figure 5 ijerph-18-01729-f005:**
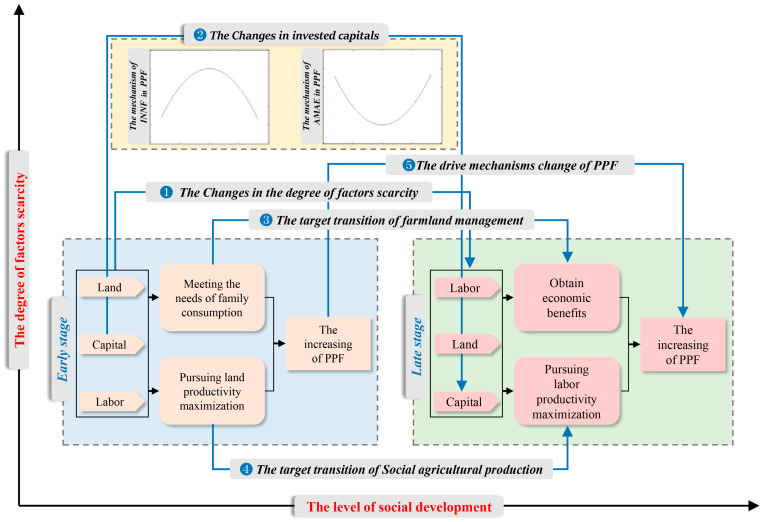
The driving mechanism of the area proportion of paddy fields in Hubei province between 1990 and 2016.

**Table 1 ijerph-18-01729-t001:** Variables and definitions.

Variable Name	Variable Type	Measurement Method	Variable Meaning	Units
Proportion of paddy field area (*PPF*)	Paddy field use changes	*PPF* = *PFA*×100%/*FA* *PFA*, paddy field area; *FA*, farmland area	Type of farmland use	%
Agricultural machinery per capita of agricultural employees (*AMAE*)	The substitution of machinery for labor	*AMAE*= *TPAM*/*AER* *TPAM*, total power of agriculture machinery; *AER*, agricultural employees in rural areas	The substitution of machinery for labor	Kw
Quadratic term of *AMAE* (*AMAE*^2^)		—		—
The input intensity of nitrogen fertilizer per hectare farmland (*INNF*)	The substitution of nitrogen fertilizer for farmland	*INNF* = *NANF*/*FA* *NANF*, net amount of nitrogenous fertilizer input	The substitution of nitrogen fertilizer for farmland	kg/ha
Quadratic term of *INNF* (*INNF*^2^)		—		—
Average annual temperature (*AT*)	Climate change	—	Temperature change	°C
Average annual precipitation (*AP*)		—	Rainfall change	mm
Land productivity (*LP*)	Agricultural production conditions	*LP* = *QGO*×100%/*GSA* *QGO*, quantity of grain output; *GSA*, grain sown area	Land quality level	kg/ha
Proportion of effectively irrigated farmland area (*PEIF*)		*PEIF* = *EIA*×100%/*FA* *EIA*, effective irrigated area;	The reliability of water resources for farmland use	%
Crop diversity (*CD*)	Risk awareness of farmers	*CD* = 1−$\overset{n}{\underset{i=1}{\sum }}p_{i}^{2}$; *P*= *SASC*/*TSAC* *SASC*, the proportion of sown area of specific crop; *TSAC*, total sown area of crops	Risk awareness of farmers	—
Proportion of cash crop sown area (*PCC*)	Socioeconomic conditions	*PCC* = *SACC*×100%/*TSAC* *SACC*, sown area of cash crops; *TSAC*, total sown area of crops	Degree of market-induced crop substitution	%
Income inequality (*II*)		*II* = *GDPPC*/*PI* *GDPPC*, *GDP* per capita; *PI*, peasants’ income	Social income gap	—
Proportion of the number of non-agricultural employees (*PNAE*)		*PNAE* = *NNAE*×100%/*FA* *NNAE*, number of non-agricultural employees	Non-agricultural employment level	

**Table 2 ijerph-18-01729-t002:** Statistical description of the variables.

Variable	Mean	Median	Maximum	Minimum	Std.	Sum
*PPF*	52.90	60.36	83.37	0.16	21.26	24,283.29
*AMAE*	2.57	1.71	13.02	0.20	2.29	1178.73
*AMAE^2^*	11.84	2.92	169.55	0.04	23.40	5436.41
*INNF*	397.32	364.99	1647.14	14.46	209.01	182,371.80
*INNF^2^*	201,457.90	133,219.60	2,713,084.00	209.19	292,442.30	92,469,158.00
*AT*	16.49	16.75	18.70	11.40	1.31	7570.24
*AP*	1174.09	1135.67	2196.76	603.80	295.48	538,904.80
*LP*	5481.34	5682.05	9356.32	1038.08	1448.15	2,515,933.00
*PEIF*	72.87	81.87	137.67	6.17	27.80	33,448.66
*CD*	0.85	0.86	0.91	0.68	0.04	391.24
*PCC*	37.91	38.85	66.12	11.15	10.80	17,399.30
*II*	3.29	2.90	7.65	1.12	1.33	1509.94
*PNAE*	41.12	41.47	77.55	0.37	16.43	18,872.16

**Table 3 ijerph-18-01729-t003:** Results of the estimations and tests of the models.

	Coef.	Std. Err.	t	*p*	95% Conf. Interval	
*LogAT*	0.146	0.401	0.360	0.716	−0.642	0.934
*LogAP*	−0.024	0.042	−0.560	0.574	−0.106	0.059
*LogLP*	0.103	0.062	1.660	0.098	−0.019	0.224
*LogPEIF*	0.244	0.049	4.980	0.000	0.147	0.340
*LogCD*	0.967	0.313	3.090	0.002	0.353	1.582
*LogPCC*	−0.140	0.051	−2.730	0.007	−0.240	−0.039
*LogII*	−0.142	0.040	−3.590	0.000	−0.220	−0.064
*LogPNAE*	−0.110	0.027	−4.010	0.000	−0.164	−0.056
*LogAMAE*	−0.002	0.034	−0.060	0.955	−0.069	0.065
(*LogAMAE*)*^2^*	0.026	0.009	2.950	0.003	0.009	0.043
*LogINNF*	0.405	0.131	3.090	0.002	0.147	0.663
(*LogINNF*)*^2^*	−0.029	0.011	−2.630	0.009	−0.051	−0.007
Constant	1.581	1.303	1.210	0.226	−0.981	4.142
Time fix	yes	Cross fix	yes		R-squared	0.989
AdjR-squared	0.988	Observations	459			

## Data Availability

No new data were created or analyzed in this study. Data sharing is not applicable to this article.

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
