# Peer review of "Impacts of Agricultural Capitalization on Regional Paddy Field Change: A Production-Factor Substitution Perspective"

_ijerph, 2021, doi:10.3390/ijerph18041729_

Round 1

Reviewer 1 Report

Dear Authors,

Please find comments atached.

Regards

Reviewer 2 Report

Authors,

It is necessary that Authors use data for explain their main objective.
Some results must be scientifically improved.

Line 149. I made a comment in about urbanization acceleration. 

Line 150. Please explain

Line 166. Explain about variable crops and credit

Line 169. Crops diversity must be define.

Line 321. Term definition

Line 363. Please explain Figure 4

Reviewer 3 Report

The title of the paper clearly reflect its contents and keywords demonstrate the significance of the research. The topic of paddy fields in China is of great importance for ensuring food security and environmental sustainability.

The abstract is sufficiently informative, covering almost all sections: purpose, methodological approach, results, and conclusions. Some recommendations/practical implications/limitations should be added.

The authors provide a summary of the current research literature (about paddy field use changes in Southeast Asia) to provide the context. In-text citations are missing between lines 53-57.

In the introduction part of the article, the authors mentioned that there are few studies on the driving mechanisms of regional paddy field use changes based on the perspective of production-factor substitution. Therefore, the present study is completely justified.

The research question and the hypothesis are missing. These should be formulated by the authors and inserted at the end of introduction part.

In-text citations are missing in the section entitled „Study area”.

There is no source mentionned for figure 1.

The authors should be more carreful to the citations in the text: e.g. “(http://www.resdc.cn/).

Please read carrefully the instructions for authors: “In the text, reference numbers should be placed in square brackets [ ], and placed before the punctuation; for example [1], [1–3] or [1,3].

The statistical analysis is appropriate, but it should be more explained in the methodology part.

The results are clearly presented and appropriately analysed.

The paper clearly identify practical implications for the rural society. It could have an economic, social and environmental impact thus improving the quality of life.

I suggest stronger links with the findings in the literature (in the discussion section). The authors should mention some aspects about the limitation of the study.

Round 2

Reviewer 1 Report

Dear authors,

Thank you for taking my comments into account.

Regards

Reviewer 3 Report

Dear authors,

Your paper was significantly improved according to reviewers suggestions. Congratulations for your work! It is a very interesting paper  with practical implications for the local economy.